# MVP: Memory-enhanced Vision-Language-Action Policy with Feedback Learning

## Abstract

Recent advances in Vision-Language-Action (VLA) models have enabled robots to perform a wide range of manipulation tasks conditioned on language instructions, offering strong generalisation across tasks, objects, and environments. However, most existing VLAs operate under a Markov assumption, limiting their ability to handle temporally extended tasks and learn from feedback. To address these limitations, we propose MVP, a non-Markovian VLA model that leverages episodic memory composed of historical actions and visual observations. To mitigate the computational cost of storing high-dimensional histories, we introduce a compact memory representation inspired by video understanding techniques. Additionally, to prevent the model from disregarding historical inputs during training, we design a novel feedback learning strategy based on SO(3) trajectory perturbation. This approach encourages the model to associate actions with their environmental consequences through observation-action-observation sequences. Experimental results on both simulated and real-world benchmarks demonstrate that MVP outperforms existing models, particularly on tasks that require temporal reasoning and history-dependent decision-making. Our findings highlight the importance of memory and feedback in advancing the capabilities of general-purpose robotic manipulation systems.

## 1 Introduction

Achieving general-purpose robotic manipulation represents a critical milestone towards the development of broadly applicable and adaptable robotic systems (Liu et al., 2024b). Recent advances (Brohan et al., 2023b;a; Driess et al., 2023; Kim et al., 2024; Li et al., 2024b) have introduced Vision-Language-Action (VLA) models, a promising paradigm that enables robots to execute a wide range of manipulation tasks conditioned on natural language instructions. VLAs offer significant advantages over traditional approaches (Chi et al., 2023; Florence et al., 2022; Jarrett et al., 2020), exhibiting strong generalisation across diverse robotic platforms, task structures, object appearances, and environmental contexts. These models typically employ an end-to-end architecture, mapping visual observations and textual commands directly to low-level executable actions. Most VLAs leverage Multimodal Large Language Models (MLLMs) (Liu et al., 2023a; Awadalla et al., 2023), which are pretrained on large-scale vision-language corpora and possess foundational multimodal reasoning capabilities that are transferable to robotic control.

Despite these advances, most VLAs exhibit a fundamental limitation: they rely on Markovian policies, predicting actions solely based on the current observation. While this Markovian formulation suffices for simple, short-horizon tasks, it fundamentally restricts the robot's ability to reason about temporally extended goals or to learn from the consequences of previous actions. Markovian policies are inherently incapable of solving tasks that require historical information, as they lack temporal memory. Consequently, they fail in settings where success depends on prior states or actions. For example, to exchange the positions of two objects, the robot must retain information regarding the initial configuration. Similarly, executing continuous or goal-directed motions (e.g., "swipe the table") often leads to erratic or unstable trajectories when prior steps are not considered. Although hierarchical action models (Intelligence et al., 2025; Li et al., 2025b) may partially address this issue, they introduce additional model complexity, data requirements, and inference latency, which can hinder real-time deployment. Moreover, Markovian policies are unable to learn from feedback: effective closed-loop control requires models to adapt their behaviour over time by incorporating

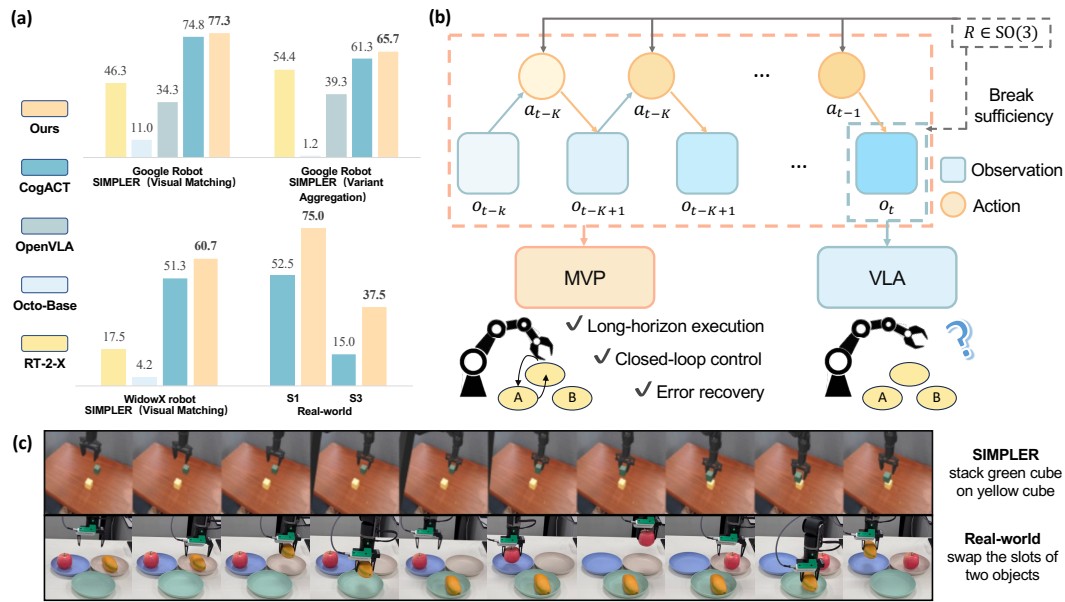

Figure 1: **Overview of MVP. (a) Performance:** MVP outperforms prior VLA methods across both simulated (SIMPLER) and real-world tasks, particularly in scenarios requiring memory and temporal reasoning. **(b) Methodology:** Unlike standard VLAs that utilise only the current observation, MVP leverages a history of observations and actions to generate temporally coherent action sequences. **(c) Task execution:** MVP successfully completes complex tasks such as stacking and object swapping by utilising past context to inform future actions.

prior commands and corresponding robot responses to refine future predictions. A memoryless model lacks the capacity to reason about cause and effect across temporal horizons, thereby limiting its adaptability and responsiveness.

To address these challenges, we propose MVP, a Vision-Language-Action model that implements a non-Markovian policy and supports learning from interaction feedback. Rather than conditioning exclusively on the current state, MVP is trained to predict actions based on an ***episodic memory comprising previous actions and 2D visual observations***. This design facilitates temporally grounded reasoning and enables feedback-aware decision making. However, integrating memory into VLAs introduces two primary challenges. First, storing raw historical image observations incurs substantial computational and memory overhead. Inspired by recent advances in video understanding (Xu et al., 2024; Zhang et al., 2024a), we compress these observations into a compact and efficient latent representation, thereby enabling scalable memory encoding. While prior works (Cheang et al., 2024; Li et al., 2024c) have explored the integration of memory information into robotic manipulation, our experiments reveal that naively expanding the input space to include history often results in models that disregard temporal information during training. This is primarily because most tasks in commonly used datasets (O'Neill et al., 2024) can be solved without explicit temporal reasoning, which encourages shortcut learning. To counteract this tendency, we introduce ***feedback learning via SO(3)***[1] ***data augmentation***. Specifically, we apply random rotations sampled from the SO(3) group to the end-effector trajectory during training. In contrast to SE(3)-equivariant approaches (Yang et al., 2024b; Tie et al., 2025), which enforce architectural invariance, our method incentivises the model to learn the dynamic correspondence between actions and their visual consequences, thereby enhancing its temporal and causal reasoning capabilities through observation-action-observation cycles.

We train MVP on a subset of the Open-X-Embodiment dataset (O'Neill et al., 2024) and conduct extensive evaluations in both simulated and real-world environments. For simulation, we employ SimplerEnv (Li et al., 2024d), a standardised benchmark for manipulation policy evaluation. Our approach achieves a $4\%$ improvement on Google Robot tasks and a $9\%$ improvement on WidowX Robot tasks. While memory is not strictly necessary for these benchmarks, MVP demonstrates increased robustness under temporally extended feedback. To further assess the role of memory, we

---

[1]SO(3) denotes the group of 3D rotation matrices, i.e., special orthogonal $3 \times 3$ matrices with determinant 1.

design real-world tasks that explicitly require temporal reasoning, such as object swapping (Chen et al., 2024). In these scenarios, MVP significantly outperforms baseline models. These results underscore the importance of integrating memory into VLA models, enabling them to perform complex, non-Markovian reasoning and to adaptively learn from interaction history.

## 2 RELATED WORKS

### 2.1 ROBOTIC MANIPULATION

The field of robot manipulation has undergone significant evolution. Early approaches relied on rule-based systems (Dillmann & Friedrich, 1996; Stengel, 1994; Paul, 1981), in which operators manually guided robots to record trajectories or employed precise model-based control. Subsequently, imitation learning and reinforcement learning (Ho & Ermon, 2016; Reddy et al., 2020; Lu et al., 2023) introduced data-driven methodologies for robotic manipulation. Diffusion policy methods and their successors (Chi et al., 2023; Chen et al., 2024; Ze et al., 2024) have further advanced robotic action generation by leveraging the generative capabilities of the denoising diffusion process (Ho et al., 2020; Song et al., 2021a;b). Building upon these foundations, works such as Act3D (Gervet et al., 2023), PerAct (Shridhar et al., 2022; Grotz et al., 2024), GNFactor (Ze et al., 2023), RVT (Goyal et al., 2023), and SAM2Act (Fang et al., 2025) have further investigated imitation learning for robotic manipulation. Inspired by recent advances in multimodal large language models (MLLMs), approaches like VoxPoser (Huang et al., 2023) and ReKep (Huang et al., 2024) utilize MLLMs to generate constraints for optimization-based trajectory planning. Notably, SE(3)-equivariant policies (Tie et al., 2025; Yang et al., 2024b) have also provided significant inspiration.

### 2.2 VISION-LANGUAGE-ACTION MODELS

Vision-Language Models (VLMs) (Li et al., 2023; Radford et al., 2021; Awadalla et al., 2023; Karamcheti et al., 2024; Liu et al., 2023a) have achieved significant success in integrating visual and linguistic information, enabling tasks such as image captioning and visual question answering. Recent studies (Brohan et al., 2023b;a; Driess et al., 2023) have extended these models by fine-tuning pretrained VLMs to directly predict robotic actions in an end-to-end fashion. Subsequent works have introduced notable advancements. For instance, $\pi_0$ (Black et al., 2025) integrates a pretrained vision-language model with flow matching for action generation, supporting precise dexterous manipulation across multiple robot embodiments in complex tasks such as laundry folding. OpenVLA (Kim et al., 2024) proposes a 7B-parameter open-source vision-language-action model that surpasses larger proprietary models and enables efficient fine-tuning on consumer hardware for robotic manipulation. $\pi_{0.5}$ (Intelligence et al., 2025) further extends generalization to novel home environments for complex tasks via hierarchical task planning. Other improvements include enhanced modelling (Li et al., 2024b; Zhang et al., 2025a; Liu et al., 2025; Kim et al., 2025; Li et al., 2024c; Qu et al., 2025; Cheang et al., 2024; Wu et al., 2024), optimization of inference speed (Yue et al., 2024; Zhang et al., 2025b), and explorations of reasoning and reinforcement learning (Guo et al., 2025; Zhang et al., 2024b; Zhao et al., 2025; Michał et al., 2024). Despite the promising performance of these methods, most predominantly employ a Markovian formulation. Although GR-2 (Cheang et al., 2024) and RoboVLM (Li et al., 2024c) utilize historical images and states (or actions) for action prediction, our approach seeks to further improve the computational efficiency of leveraging historical information by introducing a feedback learning training strategy.

### 2.3 MEMORY IN VIDEO UNDERSTANDING

The rapid advancements in Large Language Models (LLMs) (Brown et al., 2020; Ouyang et al., 2022; Touvron et al., 2023a;b; Dubey et al., 2024; Yang et al., 2024a) and Multimodal Large Language Models (MLLMs) (Li et al., 2023; Dai et al., 2024; Liu et al., 2023a; 2024a; Li et al., 2024a) have paved the way for the development of Video Language Models. In video understanding tasks, a central challenge lies in effectively representing videos through efficient and compact token embeddings. For instance, LLaMA-VID (Li et al., 2025a) encodes per-frame features using just two tokens. Some works (Ye et al., 2025; 2024) introduce novel token compression methods to alleviate computational burdens. Vista-LLaMA (Ma et al., 2024) adopts a sequential visual projector to compress an entire video into a smaller token representation. Chat-UniVi (Jin et al., 2024) leverages dynamic tokens to

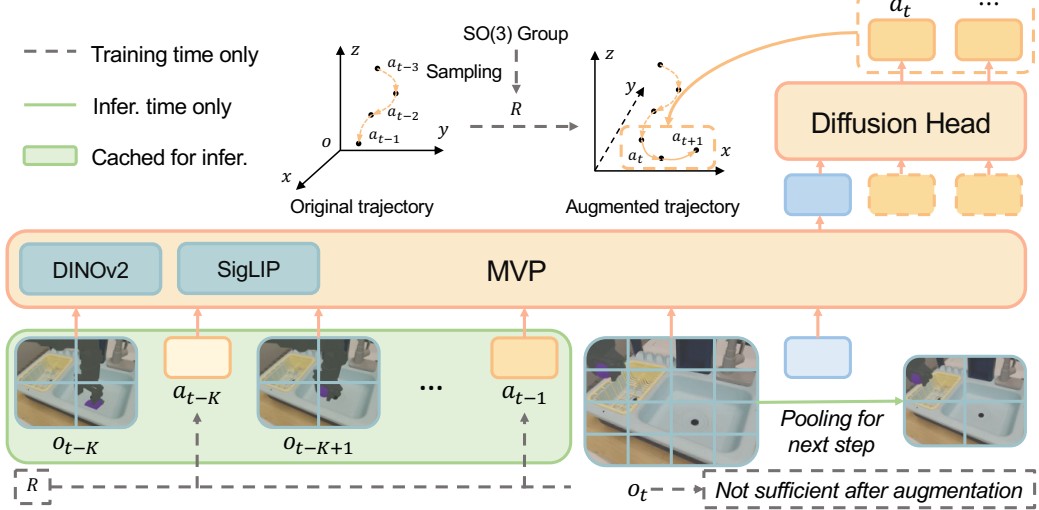

Figure 2: **Pipeline of the proposed MVP.** The MVP framework processes a sequence of historical observations and actions, together with a language instruction, to predict future action chunks. During training, SO(3) augmentation is applied to action trajectories, compelling the model to utilize historical context. Visual and action features are encoded and pooled before being processed by MVP and a diffusion head for action generation. Grey dashed and green elements indicate training-only and inference-specific components, respectively.

represent image and video features across varying scales. Recent works Flash-VStream (Zhang et al., 2024a; Wang et al., 2024) and PLLaVA (Xu et al., 2024) explore memory-based pooling strategies for more efficient video understanding, excelling in both online and offline video scenarios.

## 3 METHODOLOGY

In this section, we present the MVP algorithm, a memory-augmented Vision-Language-Action model. Section 3.1 provides the problem formulation, followed by detailed discussions of the feedback learning strategy for leveraging historical information and the overall model architecture in Sections 3.2 and 3.3, respectively.

### 3.1 PROBLEM FORMULATION

**Markovian Policy.** A significant body of work in VLA modelling frames the robot control problem as a Markov Decision Process (MDP) (Brohan et al., 2023b;a; Octo Model Team et al., 2024; O'Neill et al., 2024; Kim et al., 2024; Li et al., 2024b). In this formulation, a policy $\pi$ maps the current state $s_t$ and a language instruction $l$ to a sequence of future actions. The state $s_t$ encapsulates the agent's perception of the environment, typically comprising visual observations and proprioceptive information (e.g., robot joint states). The policy is thus defined as:

$$\pi : (l, s_t) \rightarrow a_{t:t+N}, \tag{1}$$

where $a_{t:t+N}$ is a chunk of $N$ future actions (Zhao et al., 2023). A prevalent approach defines the action space in terms of the robot's end-effector. A common action parametrization is:

$$a_t = [\Delta x_t, \Delta \theta_t, g_t], \tag{2}$$

where $\Delta x_t \in \mathbb{R}^3$ denotes the relative translation of the end-effector, $\Delta \theta_t \in \mathbb{R}^3$ represents the rotational change, and $g_t \in \{0, 1\}$ indicates the gripper state (open/closed).

**Non-Markovian Policy.** To enable reasoning over longer time horizons and address complex, temporally-extended tasks, other approaches formulate the problem as a non-Markov Decision Process (nMDP) (Li et al., 2024c; Cheang et al., 2024). The policy is conditioned on the history $h_t$:

$$\pi : (l, h_t) \rightarrow a_{t:t+N}, \quad \text{where} \quad h_t = (s_0, a_0, s_1, a_1, \ldots, a_{t-1}, s_t). \tag{3}$$

While an nMDP can be converted to an MDP by augmenting the state to include the entire history $h_t$ (Qin et al., 2023; Toro Icarte et al., 2019), this method has practical drawbacks. State augmentation can lead to an intractably high-dimensional state space and, more importantly, treats the history as a monolithic entity, failing to explicitly model the rich temporal structure within the sequence. Therefore, directly formulating the problem as an nMDP is often more effective, as it allows for the use of sequence-aware architectures, such as Transformers, that are inherently designed to process and reason over historical context.

## 3.2 FEEDBACK LEARNING

Most tasks (e.g., placing an object, closing a door, or turning on a device) in current robot manipulation datasets (O'Neill et al., 2024; Brohan et al., 2023b; Walke et al., 2023) can be addressed using a MDP, which accounts for the strong performance of Markovian policies. However, this data bias poses a unique challenge for non-Markovian policies.

**Theorem.** For a MDP, the current state $s_t$ is a sufficient statistic for its history $h_t$. In other words,

$$P(s_{t+1} \mid h_t, a_t) = P(s_{t+1} \mid s_t, a_t), \tag{4}$$

which implies that *the policy $\pi(a_t \mid h_t)$ is equivalent to $\pi(a_t \mid s_t)$ under the MDP assumption*. Please refer to the supplementary material for a detailed proof.

Although manipulation tasks may not strictly satisfy the Markov property in practice, this result nonetheless suggests a shortcut for non-Markovian policy models—namely, predicting the action solely based on the current state. Our experiments, which visualise the model's attention maps, corroborate the existence of this shortcut (see Figure 5 for details).

A straightforward remedy is to train the model exclusively on history-dependent tasks. However, this approach is impractical due to the significant data collection costs. To address this, we introduce a time-invariant SO(3) distortion to the end-effector trajectory, i.e., predicting robot actions in a randomly selected coordinate frame rather than a fixed one:

$$\boldsymbol{a}_t = [\text{apply\_rotation}(\Delta\boldsymbol{x}_t, \Delta\boldsymbol{\theta}_t, R), g_t], \quad \text{for } t = 0, 1, 2, \dots \tag{5}$$

where $R$ is a rotation matrix randomly sampled from the SO(3) group. *By introducing this distortion into the historical trajectory, the current state ceases to be a sufficient statistic for $h_t$ since it no longer encodes the distortion information.* Consequently, the model must infer the distortion from the history to yield an optimal policy. This process is akin to learning from observation-action-observation pairs to infer the environmental feedback corresponding to a particular action. We thus refer to this as the ***feedback learning*** strategy.

## 3.3 MODEL DESIGN

Given a robot trajectory $(\boldsymbol{s}_{t-K}, \boldsymbol{a}_{t-K}, \dots, \boldsymbol{a}_{t-1}, \boldsymbol{s}_t)$ as input, we use the visual observation $\boldsymbol{o}_t$ as a proxy for the state $\boldsymbol{s}_t$. Features are extracted from this trajectory by processing the visual observations with DINOv2 (Oquab et al., 2024) and SigLIP (Zhai et al., 2023), and the actions with a multi-layer perceptron (MLP):

$$f_{\boldsymbol{o},\tau} = \text{MLP}(\text{Concat}(\text{DINOv2}(\boldsymbol{o}_\tau), \text{SigLIP}(\boldsymbol{o}_\tau))), \tag{6}$$

$$f_{\boldsymbol{a},\tau} = \text{MLP}(\boldsymbol{a}_\tau), \tag{7}$$

$$\text{for } \tau = t - K, t - K + 1, \dots, t. \tag{8}$$

Since VLA models are designed for robotic control, inference speed is critical, and excessively long context windows are impractical. To mitigate this issue, we employ adaptive pooling for historical observations. To facilitate action chunk generation, we follow CogACT (Li et al., 2024b), first generating an action feature using a large language model (LLM):

$$f_t = \text{LLM}(\boldsymbol{l}, \text{AP}(f_{\boldsymbol{o},t-K}), f_{\boldsymbol{a},t-K}, \text{AP}(f_{\boldsymbol{o},t-K+1}), \dots, f_{\boldsymbol{a},t-1}, f_{\boldsymbol{o},t}), \tag{9}$$

where AP denotes adaptive pooling. Note that no pooling is applied to the current observation in order to preserve maximal visual information. Our experiments demonstrate that by compressing historical information, the proposed method can accommodate histories exceeding 128 steps within 24 GB of memory, while maintaining acceptable inference speed.

Table 1: Comparison of our approach with existing VLA models on the Google robot across four tasks in two SIMPLER (Li et al., 2024d) settings. All models are trained on the Open X-Embodiment dataset, except for RT-1 which is trained exclusively on the Google robot subset. The results of other methods are from (Li et al., 2024b).

| | Visual Matching | | | | | Variant Aggregation | | | | |
|---|---|---|---|---|---|---|---|---|---|---|
| Method | Pick Coke Can | Move Near | Open/Close Drawer | Open and Place | *Avg* | Pick Coke Can | Move Near | Open/Close Drawer | Open and Place | *Avg* |
| RT-1 | 85.7 | 44.2 | 73.0 | 6.5 | 52.4 | 89.8 | 50.0 | 32.3 | 2.6 | 43.7 |
| RT-1-X | 56.7 | 31.7 | 59.7 | 21.3 | 42.4 | 49.0 | 32.3 | 29.4 | 10.1 | 30.2 |
| RT-2-X | 78.7 | 77.9 | 25.0 | 3.7 | 46.3 | 82.3 | 79.2 | 35.3 | 20.6 | 54.4 |
| Octo-Base | 17.0 | 4.2 | 22.7 | 0.0 | 11.0 | 0.6 | 3.1 | 1.1 | 0.0 | 1.2 |
| OpenVLA | 18.0 | 56.3 | 63.0 | 0.0 | 34.3 | 60.8 | 67.7 | 28.8 | 0.0 | 39.3 |
| RoboVLMs | 77.7 | 62.9 | 42.6 | 23.1 | 51.6 | 50.2 | 62.5 | 33.1 | 23.3 | 42.3 |
| CogACT | 91.3 | **85.0** | 71.8 | 50.9 | 74.8 | 89.6 | **80.8** | 28.3 | 46.6 | 61.3 |
| Ours | **95.7** | 80.8 | **75.0** | **57.4** | **77.2** | **90.1** | 72.7 | **42.3** | **57.7** | **65.7** |

Table 2: Evaluation results on the WidowX robot in the SIMPLER *Visual Matching* setting. For these tests, we repeat each task 5 times to improve the statistical significance. The results of other methods are from (Li et al., 2024b).

| Method | Put Spoon on Towel | Put Carrot on Plate | Stack Green Block on Yellow Block | Put Eggplant in Yellow Basket | *Avg* |
|---|---|---|---|---|---|
| RT-1-X | 0.0 | 4.2 | 0.0 | 0.0 | 1.1 |
| Octo-Base | 15.8 | 12.5 | 0.0 | 41.7 | 17.5 |
| Octo-Small | 41.7 | 8.2 | 0.0 | 56.7 | 26.7 |
| OpenVLA | 4.2 | 0.0 | 0.0 | 12.5 | 4.2 |
| RoboVLMs | 45.8 | 25.0 | 7.5 | 78.3 | 39.2 |
| CogACT | 71.7 | 50.8 | 15.0 | 67.5 | 51.3 |
| Ours | **72.5** | **64.2** | **16.7** | **89.2** | **60.7** |

Subsequently, we employ a DiT (Peebles & Xie, 2023) model to generate the action chunk conditioned on $f_t$, using the DDIM (Song et al., 2021a) reverse denoising process:

$$\hat{a}_{t:t+N}^{\tau-1} = \sqrt{\alpha^{\tau-1}} \left( \frac{\tilde{a}_{t:t+N}^{\tau} - \sqrt{1-\alpha^{\tau}} \cdot \epsilon_\theta(\tilde{a}_{t:t+N}^{\tau}, \tau, f_t)}{\sqrt{\alpha^{\tau}}} \right) + \sqrt{1-\alpha^{\tau-1}} \cdot \epsilon_\theta(\tilde{a}_{t:t+N}^{\tau}, \tau, f_t),$$
(10)

where $\tilde{a}_{t:t+N}^{\tau}$ is the noisy sample at diffusion timestep $\tau$, $\alpha^{\tau}$ is the noise schedule, and $\epsilon_\theta$ is the DiT denoiser predicting the added noise, conditioned on $f_t$.

The model is trained by minimising the standard noise prediction loss:

$$\mathcal{L}_{\text{diff}} = \mathbb{E}_{a_{t:t+N}^0, \epsilon, \tau} \left[ \left\| \epsilon - \epsilon_\theta(\tilde{a}_{t:t+N}^{\tau}, \tau, f_t) \right\|_2^2 \right],$$
(11)

where $\tilde{a}_{t:t+N}^{\tau} = \sqrt{\alpha^{\tau}} a_{t:t+N}^0 + \sqrt{1-\alpha^{\tau}} \epsilon$ is the noised version of the ground-truth action chunk $a_{t:t+N}^0$, and $\epsilon \sim \mathcal{N}(0, \mathbf{I})$ is standard Gaussian noise.

## 4 EXPERIMENTS

### 4.1 IMPLEMENTATION DETAILS

**Training Dataset.** We utilize the Open X-Embodiment (OXE) (O'Neill et al., 2024) dataset as the primary source for training. This dataset comprises over one million real-world robotic trajectories aggregated from 60 individual datasets, encompassing 22 distinct robotic embodiments. Consistent with prior works (Octo Model Team et al., 2024; Kim et al., 2024; Li et al., 2024b), we employ a similar subset of OXE for training, which consists of 22.5 million frames. Further information regarding data distributions can be found in the supplementary materials.

**Model Details.** The model training follows a two-stage pipeline. During the pretraining stage, the primary focus is to enable the model to process historical inputs effectively. Model initialization leverages pre-trained vision and language modules from (Li et al., 2024b), and all modules are trained end-to-end. In the fine-tuning stage, we employ a batch size of 16 and a memory length of 128.

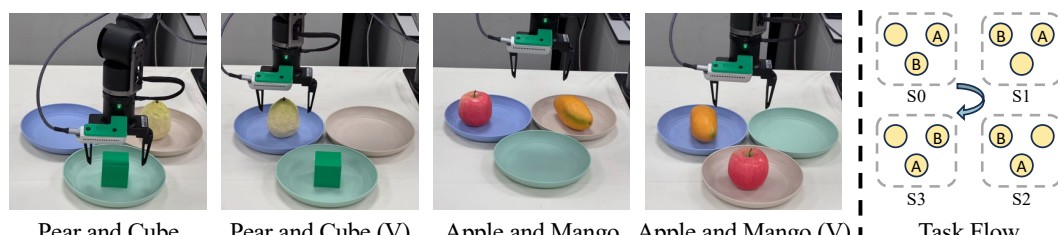

| Pear and Cube | Pear and Cube (V) | Apple and Mango | Apple and Mango (V) | Task Flow |

Figure 3: **Illustration of the swapping objects task. Left:** Example images of the four task variants used in our experiments, showing different object pairs (i.e., "Pear and Cube" and "Apple and Mango") and their initial placements. The robot is required to utilise the empty plate to successfully swap the positions of the two objects. **Right:** The task flow diagram depicts the sequential stages required to complete the swapping objects task, beginning from the initial state (**S0**) and proceeding through intermediate states (**S1**, **S2**) to the final state (**S3**).

This stage is performed exclusively on real-world data requiring temporal memory. All training experiments are conducted on eight 80GB GPUs. For more details, please see the Appendix A.3.

## 4.2 SIMULATION RESULTS

We evaluate MVP using the SIMPLER (Li et al., 2024d) benchmark, which provides reproducible, near-photorealistic robotic manipulation environments. Our experiments utilize two SIMPLER settings: *Visual Matching* for sim-to-real visual alignment and *Variant Aggregation* to test robustness against randomized visual elements.

**Google Robot (Table 1).** MVP outperforms all baselines, achieving the highest average success rates in both *Visual Matching* (77.2%) and *Variant Aggregation* (65.7%) settings.

**WidowX Platform (Table 2).** Here, MVP sets a new state-of-the-art with a 60.7% average success rate, a 9.4% improvement over the previous best, demonstrating robust performance across all tasks.

Across both platforms, MVP consistently surpasses existing VLA models, even though these tasks do not strictly require temporal memory. This performance gain stems from our effective feedback learning strategy, which leverages observation-action-observation sequences and SO(3) trajectory perturbations. These results confirm that robust feedback learning significantly enhances VLA policy performance, even in scenarios without inherent memory demands.

## 4.3 REAL-WORLD DEPLOYMENT

To evaluate the temporal reasoning capabilities of the models, we design an experimental setting in which the current observation does not provide historical information and is insufficient for making future decisions. Although CALVIN (Mees et al., 2022) introduces tasks emphasising long-horizon execution, its evaluation framework decomposes tasks into several subtasks, and Markovian policies (Zhang et al., 2025a) have demonstrated strong performance on the CALVIN dataset. Consequently, CALVIN does not fully satisfy our requirements for assessing memory-based decision-making.

To address this limitation, we collect a real-world dataset specifically designed to evaluate memory-dependent decision-making. In particular, we adopt the swapping objects task (Chen et al., 2024): three plates are arranged on a table, with two of them each holding an object. The objective is to use the empty plate to swap the positions of the two objects. We design four variants with different object types (i.e., "apple and mango", "pear and cube") and initial placements to facilitate a more generalizable evaluation. Further details are provided in Figure 3. For each variant, we collect 40 demonstration trajectories for training.

As shown in Figure 3, the task involves a sequence of three pick-and-place steps (S1–S3). Critically, the visual observation at any given time is ambiguous regarding the current task stage, making memory of past actions essential for success. We evaluate each variant over 10 trials using a 6-DoF GALAXEA A1 robot. Please refer to Appendix A.1 for more details about the setup.

Table 3: **Step-wise success rates of the swapping objects task under four real-world variants.** Each entry reports the proportion of successful trials (out of 10) in which the specified state was achieved. The final column presents the average S3 success rate across all variants.

| Method | Pear and Cube | | | Pear and Cube (V) | | | Apple and Mango | | | Apple and Mango (V) | | | Mean (S3) |
|---|---|---|---|---|---|---|---|---|---|---|---|---|---|
| | S1 | S2 | S3 | S1 | S2 | S3 | S1 | S2 | S3 | S1 | S2 | S3 | |
| CogACT | 0.6 | 0.3 | 0.1 | 0.4 | 0.3 | 0.1 | 0.5 | 0.2 | 0.2 | 0.6 | 0.2 | 0.2 | 0.150 |
| Ours | **0.7** | **0.5** | **0.3** | **0.6** | **0.4** | **0.3** | **0.8** | **0.5** | **0.5** | **0.9** | **0.5** | **0.4** | **0.375** |

Table 4: **Error decomposition for real-world experiments.** The table below summarizes the frequency of each error type over 40 trials.

| **Method** | Fail to Grasp | *Wrong Pick* | *Wrong Place* | *Early Stop* | **State-dependent Errors** | **Total Errors** |
|---|---|---|---|---|---|---|
| CogACT | 9 | 5 | 9 | 11 | 25 | 34 |
| Ours | 8 | 2 | 6 | 9 | 17 | 25 |

### 4.3.1 RESULTS AND ANALYSIS

As shown in Table 3, our non-Markovian policy consistently outperforms the Markovian baseline, CogACT, across all task variants and stages. The most significant improvement is at the final goal state (S3), where our method achieves an average success rate of $0.375$, more than double CogACT's $0.150$. This highlights the advantage of memory-based policies for long-horizon tasks.

Qualitative analysis reveals that CogACT, lacking the ability to infer the current task stage, often oscillates between states. Its occasional successes are largely due to chance in tasks with overlapping state transition actions, confirming the inherent limitations of Markovian approaches for such problems. In contrast, our method effectively tracks task progress and leverages historical information to recover from errors, demonstrating robust execution and adaptability.

Furthermore, we conducted an error decomposition analysis on real-world tasks, as presented in Table 4. The results indicate that while grasping failures are comparable between the two methods, MVP substantially mitigates *state-dependent errors (wrong pick/place and early stop)*, which can only be prevented through the tracking of historical information. We believe that the combination of the real-world experiments and the error decomposition analysis convincingly demonstrates the superiority of our method in long-horizon tasks.

### 4.4 ABLATION STUDY

To assess the effectiveness of the proposed components in MVP, we conduct a comprehensive ablation study on the WidowX robot in the SIMPLER Visual Matching setting. Specifically, we investigate the impact of SO(3) augmentation and memory length on task performance. The results are summarised in Table 5.

**Effect of Memory Length.** Given that both the training and evaluation tasks possess the Markov property, increasing the memory length does not consistently yield performance improvements. As reported in Table 5, the model achieves optimal performance with a moderate memory length of 8 steps, whereas extending the memory window further leads to diminishing, or in some cases, slightly reduced performance. Nevertheless, maintaining an appropriate memory buffer enhances robustness to temporal variations, indicating that a moderate memory length remains advantageous even when the Markov assumption holds.

**Ablation of Feedback Learning.** Omitting SO(3) augmentation (i.e., the feedback learning strategy) results in inferior performance (average success rate of $53.8\%$), highlighting the importance of the proposed augmentation for non-Markovian policy learning. The inclusion of $z$-axis SO(3) augmentation with a range of $\pm 45°$ substantially enhances performance relative to models trained without augmentation. By contrast, applying full $xyz$-axis augmentation leads to a marked decrease

Table 5: **Ablation study on the WidowX robot in the SIMPLER Visual Matching setting.** We assess various configurations of our method by altering the SO(3) augmentation strategy and memory length. Here, "$z$" indicates rotation is applied solely around the $z$-axis, while "$xyz$" denotes rotation about all axes. "±45" means the augmentation angle is randomly sampled within the range -45° to +45° for each trajectory. The accompanying number specifies the augmentation range. The model with a memory length of 128 is fine-tuned from the model with a memory length of 8.

| SO(3) Aug. | Memory Length | Put Spoon on Towel | Put Carrot on Plate | Stack Green Block on Yellow Block | Put Eggplant in Yellow Basket | *Avg* |
|---|---|---|---|---|---|---|
| $z, \pm45°$ | 4 | 68.3 | 61.7 | 14.2 | 78.3 | 55.6 |
| $z, \pm45°$ | 8 | **72.5** | **64.2** | 16.7 | **89.2** | **60.7** |
| $z, \pm45°$ | 16 | 70.8 | 63.3 | 17.5 | **89.2** | 60.2 |
| $z, \pm45°$ | 128 (ft) | 69.2 | 59.2 | **19.2** | 85.8 | 58.4 |
| $xyz, \pm45°$ | 8 | 63.3 | 53.3 | 12.5 | 72.5 | 50.4 |
| – | 8 | 67.5 | 56.7 | 15.0 | 75.8 | 53.8 |

Table 6: **Inference speed comparison.** All results are obtained on a single RTX 3090 GPU.

| Method | Control frequency (w/o action chunk) | Control frequency (w/ action chunk) | GPU Memory Consumption |
|---|---|---|---|
| OpenVLA | 2.2 Hz | - | 16.6 GB |
| CogACT | 3.7 Hz | 59.2 Hz | 16.9 GB |
| Ours (8 steps) | 3.2 Hz | 51.2 Hz | 17.3 GB |
| Ours (128 steps) | 2.2 Hz | 35.2 Hz | 22.7 GB |

in performance (average success rate of $50.4\%$), indicating that excessive augmentation may overly complicate the learning problem.

## 4.5 INFERENCE SPEED

We evaluate the inference speed of our MVP model on a single RTX 3090 GPU. The results are shown in Table 6. By employing the proposed history memory compression and the action chunking technique (Zhao et al., 2023), MVP achieves a comparable inference speed (51.2 Hz) to CogACT (Li et al., 2024b) when using a memory size of 8 steps. Even with a history size of 128 steps, which suffices for most memory-dependent manipulation tasks, our model maintains an acceptable inference speed (35.2 Hz with action chunking).

## 5 CONCLUSION AND LIMITATIONS

**Conclusion.** We presented MVP, a non-Markovian Vision-Language-Action model designed to enhance robotic manipulation in tasks requiring temporal reasoning and feedback-based learning. Unlike Markovian VLA models that rely solely on the current observation, MVP integrates a history of past actions and visual inputs, facilitating context-aware policy learning. To ensure effective utilization of memory, we introduced a novel SO(3) augmentation strategy that breaks Markov sufficiency and compels the model to leverage historical information. Our experiments on both simulated benchmarks and real-world manipulation tasks demonstrate substantial improvements over existing methods, particularly in long-horizon and memory-dependent scenarios.

**Limitations.** Nevertheless, several limitations should be noted. First, our model is trained primarily on Markovian datasets within the imitation learning framework, which restricts the diversity of temporal dependencies encountered during training. This may limit the model's capacity for feedback-driven and causal reasoning. Future research could address this issue by utilizing memory-dependent task data and reinforcement learning. Second, while we use compact memory representations, further optimization of inference speed and memory efficiency is necessary for real-time deployment.

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

# A  APPENDIX

## A.1  REAL-WORLD EXPERIMENT SETUP

Our real-world experiments are conducted using the setup illustrated in Figure 4. The system is centered around a 6-DoF GALAXEA A1 robotic arm equipped with a GALAXEA G1 parallel gripper. For visual perception, the hardware includes two cameras: a static Intel RealSense L515 camera providing a third-person view of the workspace, and a wrist-mounted Intel RealSense D435i.

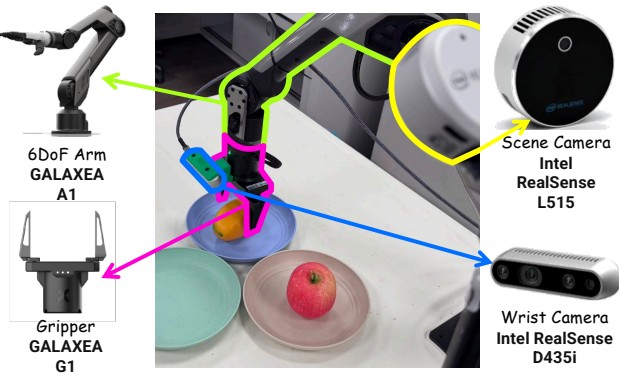

Figure 4: **Our real-world experimental setup.** The system features a 6-DoF GALAXEA A1 arm and a G1 gripper. Perception is provided by the static Intel RealSense L515 scene camera. Note that while a wrist-mounted Intel RealSense D435i camera is present on the hardware, it was not used in our experiments.

## A.2  PROOF OF THEOREM

**Theorem.** For a Markov decision process (MDP), the current state $s_t$ is a sufficient statistic for the historical trajectory $h_t = (s_0, a_0, s_1, a_1, \ldots, s_t)$. That is,

$$P(s_{t+1} \mid h_t, a_t) = P(s_{t+1} \mid s_t, a_t), \tag{12}$$

which implies that *the policy $\pi(a_t \mid h_t)$ is equivalent to $\pi(a_t \mid s_t)$ under the MDP assumption*.

**Proof:**

Consider an agent interacting with an environment modelled as a Markov Decision Process (MDP). At each time step $t$, the agent observes the state $s_t$, selects an action $a_t$, and receives a reward $r_t$. The history up to time $t$ is given by

$$h_t = (s_0, a_0, s_1, a_1, \ldots, s_{t-1}, a_{t-1}, s_t).$$

We aim to demonstrate that, under the Markov property, there exists an optimal policy that depends solely on the current state, i.e.,

$$\pi(a_t \mid h_t) = \pi(a_t \mid s_t).$$

An MDP is defined by the tuple $(\mathcal{S}, \mathcal{A}, P, R, \gamma)$, where:

- $\mathcal{S}$: state space,
- $\mathcal{A}$: action space,
- $P(s_{t+1} \mid s_t, a_t)$: transition probability,
- $R(s_t, a_t)$: reward function,
- $\gamma \in [0, 1)$: discount factor.

The Markov property asserts that, for all $t$,

$$P(s_{t+1}, r_t \mid h_t, a_t) = P(s_{t+1}, r_t \mid s_t, a_t). \tag{13}$$

Thus, the future is conditionally independent of the past, given the current state and action.

Given a policy $\pi$, the state value function is defined as

$$V^\pi(s) = \mathbb{E}_\pi\left[\sum_{k=0}^\infty \gamma^k r_{t+k} \mid s_t = s\right], \tag{14}$$

and the state-action value function is defined as

$$Q^\pi(s, a) = \mathbb{E}_\pi\left[\sum_{k=0}^\infty \gamma^k r_{t+k} \mid s_t = s, a_t = a\right]. \tag{15}$$

The objective is to find a policy $\pi^*$ that maximizes $V^\pi(s)$ for all $s$.

The probability of a trajectory $\tau = (s_0, a_0, r_0, s_1, a_1, r_1, \ldots)$ under a general policy $\pi(a_t \mid h_t)$ is

$$P(\tau) = \mu(s_0) \prod_{t=0}^\infty \pi(a_t \mid h_t) \, P(s_{t+1}, r_t \mid h_t, a_t), \tag{16}$$

where $\mu(s_0)$ is the initial state distribution. By the Markov property,

$$P(s_{t+1}, r_t \mid h_t, a_t) = P(s_{t+1}, r_t \mid s_t, a_t),$$

so

$$P(\tau) = \mu(s_0) \prod_{t=0}^\infty \pi(a_t \mid h_t) \, P(s_{t+1}, r_t \mid s_t, a_t). \tag{17}$$

For any policy $\pi(a_t \mid h_t)$, define the corresponding state-based policy

$$\tilde{\pi}(a \mid s) := P_\pi(a_t = a \mid s_t = s), \tag{18}$$

i.e., the marginal probability of selecting action $a$ in state $s$ under $\pi$.

The expected return under $\pi$ is

$$J(\pi) = \mathbb{E}_\pi\left[\sum_{t=0}^\infty \gamma^t r_t\right]. \tag{19}$$

Due to the Markov property, the future evolution depends only on the current state and subsequent actions, and not on the full history.

Therefore, for any history-dependent policy, there exists a state-based policy $\tilde{\pi}$ that induces the same state-action visitation distribution.

The Bellman optimality equation for the state value function is

$$V^*(s) = \max_a \mathbb{E}[r_t + \gamma V^*(s_{t+1}) \mid s_t = s, a_t = a],$$

where $V^*(s) = \max_\pi V^\pi(s)$, and for the state-action value function,

$$Q^*(s, a) = \mathbb{E}[r_t + \gamma \max_{a'} Q^*(s_{t+1}, a') \mid s_t = s, a_t = a],$$

where $Q^*(s, a) = \max_\pi Q^\pi(s, a)$. Both equations involve optimization over actions conditioned solely on the current state $s$.

Thus, the optimal policy can be expressed as

$$\pi^*(a \mid h_t) = \arg\max_a Q^*(s_t, a),$$

which is a function only of $s_t$.

Consequently, for any MDP, there exists an optimal policy that depends exclusively on the current state:

$$\pi(a_t \mid h_t) = \pi(a_t \mid s_t).$$

That is, the current state is a sufficient statistic for optimal control in an MDP.

Table 7: **Implementation details for pretraining and fine-tuning stages.**

| Hyperparameter | Pretraining | Fine-tuning |
|---|---|---|
| Training steps | 18 k | 20 k |
| Effective batch size | 256 | 16 |
| History length | 8 | 128 |
| Adaptive pooling size | (4, 4) | (2, 2) |
| Learning rate | $2 \times 10^{-5}$ | $2 \times 10^{-5}$ |
| Learning rate scheduler | constant | constant |
| Optimizer | AdamW | AdamW |
| Weight decay | 0.0 | 0.0 |
| Warm-up | 0 | 0 |
| Gradient clipping | 1 | 1 |

Table 8: Training data mixture.

| Dataset | Ratio |
|---|---|
| Fractal (Brohan et al., 2023b) | 27.1% |
| Kuka (Kalashnikov et al., 2018) | 14.7% |
| Bridge (Walke et al., 2023) | 15.3% |
| Taco Play (Rosete-Beas et al., 2022; Mees et al., 2023) | 3.4% |
| Jaco Play (Dass et al., 2023) | 0.6% |
| Berkeley Cable Routing (Luo et al., 2024) | 0.3% |
| Roboturk (Mandlekar et al., 2019) | 2.7% |
| Viola (Zhu et al., 2022a) | 1.1% |
| Berkeley Autolab UR5 (Chen et al.) | 1.4% |
| Toto (Zhou et al., 2023) | 2.3% |
| Stanford Hydra Dataset (Belkhale et al., 2023) | 5.1% |
| Austin Buds Dataset (Zhu et al., 2022b) | 0.2% |
| NYU Franka Play Dataset (Cui et al., 2022) | 1.0% |
| Furniture Bench Dataset (Heo et al., 2023) | 2.8% |
| UCSD Kitchen Dataset (Ge Yan & Wang, 2023) | < 0.1% |
| Austin Sailor Dataset (Nasiriany et al., 2022) | 2.5% |
| Austin Sirius Dataset (Liu et al., 2023b) | 2.0% |
| DLR EDAN Shared Control (Quere et al., 2020) | < 0.1% |
| IAMLab CMU Pickup Insert (Saxena et al., 2023) | 1.0% |
| UTAustin Mutex (Shah et al., 2023) | 2.6% |
| Berkeley Fanuc Manipulation (Zhu et al., 2023) | 0.9% |
| CMU Stretch (Mendonca et al., 2023) | 0.2% |
| BC-Z (Jang et al., 2021) | 8.6% |
| FMB Dataset (Luo et al., 2023) | 2.4% |
| DobbE (Shafiullah et al., 2023) | 1.6% |

### A.3 ADDITIONAL IMPLEMENTATION DETAILS

We present the additional architecture and training procedure details in Table 7.

**Pretraining Data.** We pretrain our model using 25 VLA datasets from Open X-Embodiment (O'Neill et al., 2024). Specifically, we utilize the subsets provided by CogACT (Li et al., 2024b), which consist of 0.4 million robot trajectories, corresponding to 22.5 million frames. The detailed data split is provided in Table 8.

**Pretraining.** During the pretraining stage, a batch size of 256 is used, realized via four steps of gradient accumulation. The maximum history length is set to 8, a relatively small value, as the primary focus at this stage is to enable the model to process historical inputs effectively while maintaining a large batch size to facilitate improved performance and faster convergence. *To expose the model to varying history lengths, a shorter history length—randomly selected between 1 and 7—is employed*

*with probability 0.2.* Model initialization leverages pre-trained vision and language modules from (Li et al., 2024b). All the modules are trained end-to-end using a constant learning rate of $2 \times 10^{-5}$ for over 12 epochs.

**Fine-tuning.** In the fine-tuning stage, we employ a batch size of 16 and a memory length of 128. Fine-tuning is performed exclusively on real-world data requiring temporal memory, as further fine-tuning on the OXE dataset does not yield additional improvements. This may be attributed to the predominance of tasks in OXE that can be addressed with a Markovian approach, as discussed in detail in Section 3.3. Additional information regarding real-world task settings is provided in Section 4.3.

**Infrastructure.** All training experiments are conducted on eight 80GB GPUs utilising PyTorch's Fully Sharded Data Parallel (FSDP) framework with BFloat16 precision and flash-attention (Dao et al., 2022; Dao, 2024).

## A.4 ATTENTION WEIGHT DISTRIBUTION

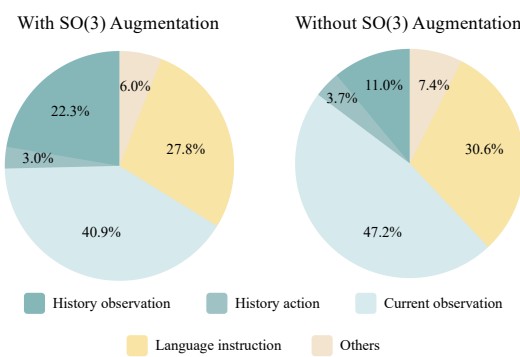

Figure 5: **Attention weight distribution.**

Figure 5 quantifies the average attention weights of the action feature $f_t$ across 20 evaluation examples. Without SO(3) augmentation, the model allocates $47.2\%$ of its attention to the current observation, with only minimal attention to historical inputs, indicating a pronounced dependence on immediate sensory information. In contrast, applying SO(3) augmentation results in a significant increase in attention directed towards historical observations and actions (from $14.7\%$ to $25.3\%$ combined), accompanied by a corresponding reduction in focus on the current observation. This redistribution of attention demonstrates that the proposed augmentation effectively encourages the model to leverage history information, thereby promoting more robust non-Markovian policy learning.

## A.5 QUALITATIVE RESULTS

Additional qualitative results are presented in Figure 6, encompassing both simulated and real-world scenarios. The top section of the figure illustrates the performance of our model within the SIMPLER (Li et al., 2024d) simulation environment. Notably, the first example demonstrates that our model is capable of recovering from errors: the robot fails to grasp the apple on the first attempt but promptly recovers and succeeds on the second attempt. The middle section depicts the deployment of our model in real-world settings, showcasing its ability to execute memory-dependent, long-horizon tasks within a single rollout. The bottom section presents a representative failure case of CogACT (Li et al., 2024b) in real-world conditions. In this scenario, the model is unable to infer the current task stage solely from the present observation, resulting in actions that merely replicate the training data. Consequently, this leads to incorrect behaviours, such as moving the object in the wrong direction.

## A.6 BROADER IMPACTS

The development of memory-augmented VLA models offers significant societal benefits. These advances may enable robots to perform more complex and adaptive tasks, thereby expanding their applicability in healthcare, elder-care, logistics, and hazardous environments. Enhanced temporal reasoning and feedback learning have the potential to improve workplace safety, increase productivity, and provide meaningful assistance to individuals with disabilities or limited mobility.

Nevertheless, such technologies also pose challenges. Increased robotic autonomy may lead to job displacement in sectors dependent on manual labour, thus exacerbating economic inequality. The capability of robots to store and utilize historical data raises concerns regarding privacy and accountability. Furthermore, increased system complexity may introduce ethical and safety risks,

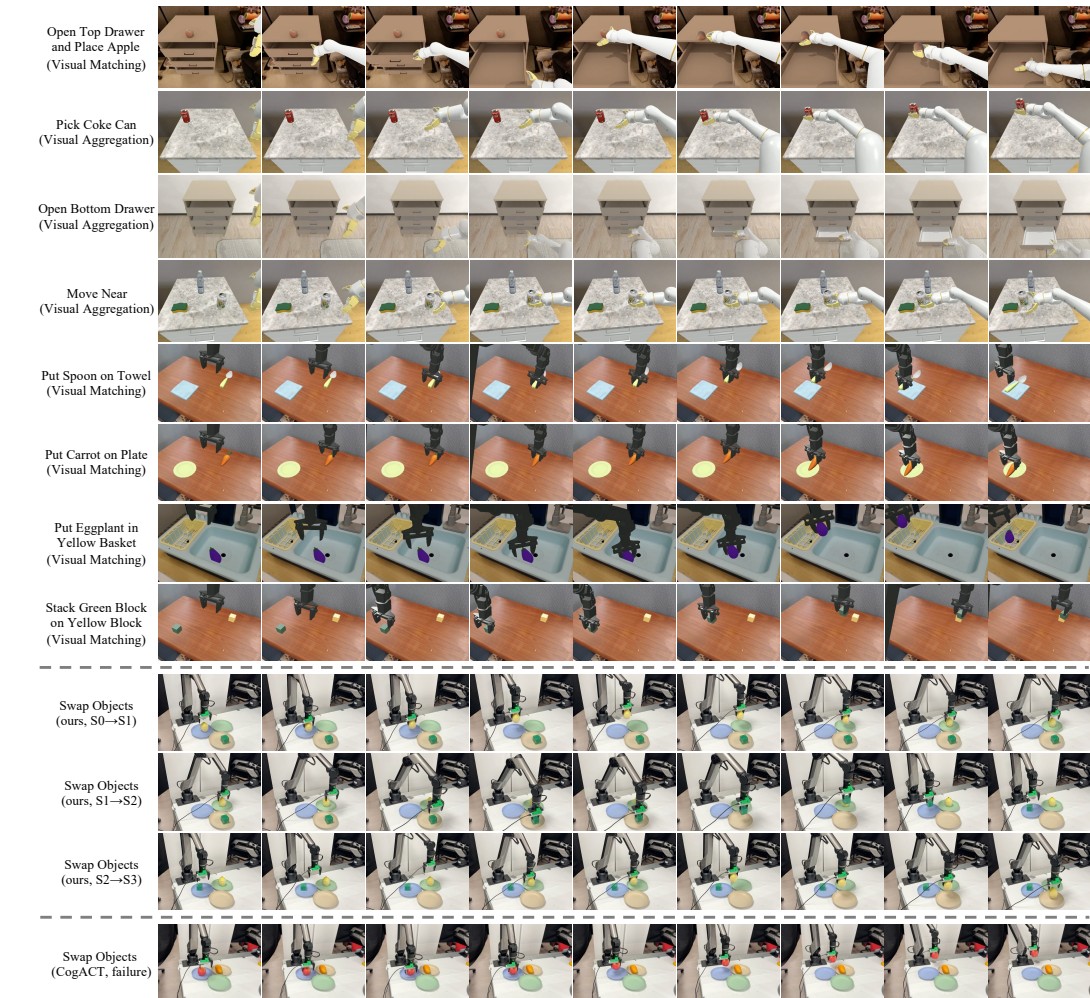

Figure 6: **Qualitative Results. Top:** Simulation results obtained using SIMPLER (Li et al., 2024d). **Middle:** Performance of our model in real-world deployment. **Bottom:** A representative failure case of CogACT (Li et al., 2024b) under our real-world setting.

such as misinterpretation of instructions or unintended behaviours. Careful consideration of these risks is essential as memory-augmented VLA models become more widely adopted.

## A.7 LLM USAGE STATEMENT

Large Language Models (LLMs) were used to assist with writing and editing portions of this paper, including improving clarity, grammar, and overall presentation of the content. All research ideas, methodology, analysis, and conclusions are entirely the work of the authors. The authors take full responsibility for all content in this paper, including any LLM-assisted text.

