# OpenReview forum: "MVP: Memory-enhanced Vision-Language-Action Policy with Feedback Learning"
_ICLR.cc/2026/Conference — ICLR 2026 Conference Withdrawn Submission_

### Official Review · Reviewer_HLxT · 2025-10-28

**Soundness:** 3
**Presentation:** 3
**Contribution:** 3
**Rating:** 6
**Confidence:** 4

**Summary:**

This paper proposes a non-Markovian Vision-Language-Action (VLA) model that incorporates past observations and actions as auxiliary information, along with an augmentation method to enhance robustness.

**Strengths:**

1. The paper is easy to understand and follow.

2. The experimental design and completeness are relatively strong.

**Weaknesses:**

1. After reading the related work section, it remains unclear how this paper differs from prior studies and what specific advancements it contributes beyond existing methods.

2. Leveraging past information is a very common strategy — many previous imitation learning approaches have incorporated historical observations [1,2]. Therefore, presenting this as a main innovation makes the contribution seem somewhat incremental. Moreover, prior works [1] and many others have also discussed the copycat problem that arises when using past observations in behavioral cloning.

[1] Fighting Copycat Agents in Behavioral Cloning from Multiple Observations. NeurIPS 2023.

[2] Rethinking Latent Redundancy in Behavior Cloning: An Information Bottleneck Approach for Robot Manipulation. ICML 2025.

**Questions:**

1. The paper states that “while a Markovian formulation suffices for simple, short-horizon tasks, it fundamentally restricts the robot’s ability to reason about temporally extended goals or to learn from the consequences of previous actions.”
 However, this limitation can be addressed by incorporating several past frames to compensate for the lack of historical information — an approach that some prior VLA models have already adopted. Therefore, the proposed method seems to mainly extend the temporal window by including additional past observations, together with the Legend data augmentation technique to improve robustness, which makes the contribution appear somewhat incremental.

2. In the introduction, the authors mention that “while memory is not strictly necessary for these benchmarks.” Could you clarify why memory is not required in the simulation benchmarks?

3. I am also confused about the use of SO(3) perturbations. Why do you introduce a distortion in the action space? Wouldn’t large perturbations make it more difficult for the model to learn the correct action mapping?

4. Finally, I am curious why setting the memory length to 8 yields the best performance. Why doesn’t increasing the amount of historical information further lead to continued improvement?

---

### Official Review · Reviewer_PQXZ · 2025-10-29

**Soundness:** 3
**Presentation:** 3
**Contribution:** 3
**Rating:** 4
**Confidence:** 3

**Summary:**

This paper introduces MVP, a robotic control model that integrates memory of past images and actions to solve complex tasks requiring long-horizon reasoning. Its core innovation is an SO(3) rotation-augmented training strategy that effectively forces the model to leverage historical information for decision-making. Experiments show MVP excels on simulated and real-world benchmarks, especially on memory-dependent tasks where traditional models fail.

**Strengths:**

-   The SO(3) augmentation strategy is a clever solution to prevent the model from "shortcut learning" and ignoring history during training.
-    The paper introduces a compact memory representation using adaptive pooling, which is highly effective, allowing the model to maintain high inference speeds and manageable memory usage, making it practical for real-world deployment.
-   The model achieves state-of-the-art performance in simulation and significantly outperforms baselines on a custom real-world task designed to be memory-dependent.

**Weaknesses:**

-   Real-world validation relies on a single task type ("object swapping"), leaving the model's generalization to other complex tasks unclear.
-    The paper convincingly shows MVP is better than a Markovian baseline (CogACT) on the memory task. However, it doesn't compare its performance against other models mentioned in the related work, such as GR-2 and RoboVLM, which also rely on historical states and images for action prediction. This omission makes it difficult to conclude that MVP's specific memory architecture and training strategy are superior to alternative designs.
-   The ablation study shows that full 3-axis SO(3) augmentation degrades performance, while z-axis only augmentation is optimal. The paper’s explanation is superficial and lacks deeper analysis. This omission leaves key questions about the method’s limitations and its true generalization capabilities unanswered.
-   While the failure modes of the baseline model are analyzed, the paper provides no corresponding analysis for the proposed MVP model, leaving its specific limitations unexamined.

**Questions:**

see weakness

---

### Official Review · Reviewer_8KjT · 2025-10-31

**Soundness:** 2
**Presentation:** 2
**Contribution:** 2
**Rating:** 2
**Confidence:** 4

**Summary:**

This paper presents MVP, a memory-enhanced Vision-Language-Action model that addresses the limitations of Markovian VLA policies by incorporating episodic memory from historical observations and actions. The key technical contributions include a compact memory representation using adaptive pooling inspired by video understanding, and a novel feedback learning strategy based on SO(3) trajectory perturbation that encourages the model to leverage temporal information during training.

**Strengths:**

1. The SO(3) augmentation approach is well-motivated—by randomly rotating trajectories, the current observation alone becomes insufficient, forcing the model to genuinely leverage historical context rather than taking shortcuts.

2. The adaptive pooling strategy for compressing historical observations is simple yet effective, enabling the model to handle long histories (128 steps) while maintaining reasonable inference speed and memory footprint.

**Weaknesses:**

1. The paper doesn't compare against other potential augmentation strategies (e.g., SE(3), temporal jittering, or other forms of trajectory perturbation). The ablation only shows z-axis vs xyz-axis rotation—this is too limited to validate the design choice. Also, applying the same rotation to the entire trajectory seems artificial; real-world disturbances are typically non-uniform.

2. The authors acknowledge that most tasks in their training data (OXE) are Markovian, yet they claim to learn non-Markovian policies. How can a model learn meaningful temporal reasoning when 90%+ of training examples don't require it? The SO(3) augmentation feels like a patch rather than a principled solution to this data problem.

3. The adaptive pooling approach is presented as inspired by "video understanding techniques," but there's no comparison with actual video compression methods (e.g., the techniques from PLLaVA, Flash-VStream that they cite). Why is (2,2) pooling optimal? What information is lost? The inference speed comparison (Table 6) shows their method is slower than CogACT, undermining claims of efficiency.

4. The paper fails to cite several highly relevant recent works on temporal modeling in VLAs, such as TraceVLA (Zheng et al., 2024) and UniVLA (Bu et al., 2025b). The comparison also lacks recent strong baselines like π0, raising questions about whether the improvements hold against state-of-the-art methods.

5. The real-world evaluation consists of only a single task type (object swapping with 4 variants), which is insufficient to demonstrate generalization of the memory mechanism. The simulated experiments use SIMPLER, which is not considered state-of-the-art, and the baseline methods (RT-2-X, OpenVLA) are relatively outdated given the rapid progress in VLA community.

6. Unclear necessity of the approach: The paper's core motivation relies on the claim that most VLA tasks require non-Markovian reasoning, yet their own results show only modest improvements (4-9%) on SIMPLER benchmarks that supposedly don't require memory. The proof in Section 3.2 merely restates the Markov property without providing compelling evidence that real manipulation tasks actually violate this assumption—the swapping task seems cherry-picked to favor their method.

**Questions:**

See weaknesses.

---

### Official Review · Reviewer_SfHz · 2025-11-04

**Soundness:** 4
**Presentation:** 3
**Contribution:** 2
**Rating:** 4
**Confidence:** 3

**Summary:**

This paper proposes MVP to overcome a fundamental weakness in current robotic manipulation models: they only consider the present state and cannot reason about temporal sequences. The authors argue that simply adding memory to these models is insufficient because they learn to ignore it, since most training tasks can be solved without historical context. Their main technical insight is to apply random rotations to robot trajectories during training, which breaks this dependency on current observations alone and forces the model to leverage past information. They compress the visual history using pooling techniques to maintain computational tractability. The experimental results reveal a modest 4-9% improvement on standard simulation benchmarks where memory is not essential, but a more convincing 2.5x performance gain on real-world tasks that genuinely require temporal reasoning. The work demonstrates that memory matters for complex manipulation, though it remains constrained by the scarcity of memory-dependent examples in existing training datasets.

**Strengths:**

The paper makes solid contributions through its SO(3) trajectory augmentation strategy, which cleverly addresses the training shortcut problem where models ignore historical information. The theoretical motivation is sound, with clear explanation of why Markovian shortcuts emerge. The experimental design appropriately combines standardized benchmarks with custom memory-dependent tasks, and the 2.5x improvement on object swapping provides convincing evidence that memory matters for long-horizon manipulation. The presentation is clear with effective visualizations, and the problem formulation distinguishing MDPs from non-MDPs is both precise and accessible.

**Weaknesses:**

The paper has two critical limitations. First, the real-world evaluation is insufficient, testing only a single task type (object swapping) with 4 variants and 40 demonstrations each, which provides weak evidence for generalization to diverse memory-dependent manipulation scenarios. Second, the absence of direct comparisons with memory-augmented baselines like GR-2 and RoboVLM makes it impossible to determine whether performance gains stem from the proposed SO(3) augmentation strategy or simply from incorporating memory itself, leaving the core contribution unclear.

**Questions:**

1. Your theorem shows current state is sufficient for MDPs, and SO(3) augmentation breaks this sufficiency. However, can you provide empirical analysis of how the model learns to use history? Specifically, does it learn causal relationships between actions and state changes, or does it simply memorize rotation-specific trajectories?

2.  Your method requires storing and processing historical observations, which becomes prohibitive for very long horizons. Have you explored hierarchical memory architectures where recent history is detailed but distant history is compressed into higher-level state summaries?

---

### Note · Authors · 2025-12-14

I have read and agree with the venue's withdrawal policy on behalf of myself and my co-authors.